# Psychological risk indicators of disordered eating in athletes

**Hannah Stoyel** [1]*, **Vaithehy Shanmuganathan-Felton**[2], **Caroline Meyer**[3], **Lucy Serpell**[1,4]

1 University College London, London, United Kingdom, 2 University of Roehampton, London, United Kingdom, 3 WMG and Warwick Medical School, University of Warwick, Coventry and Warwickshire NHS Partnership Trust, Coventry, United Kingdom, 4 North East London NHS Foundation Trust, London, United Kingdom

* Hannah.stoyel.13@ucl.ac.uk

## Abstract

### Objectives

This project examined risk factors of disordered eating in athletes by adapting and applying a theoretical model. It tested a previously proposed theoretical model and explored the utility of a newly formed model within an athletic population across gender, age, and sport type to explain disordered eating.

### Design

The design was cross-sectional and the first phase in a series of longitudinal studies.

### Methods

1,017 athletes completed online questionnaires related to social pressures, internalisation, body dissatisfaction, negative affect, restriction, and bulimia. Structural equation modelling was employed to analyse the fit of the measurement and structural models and to do invariance testing.

### Results

The original theoretical model failed to achieve acceptable goodness of fit ($\chi^2$ [70, 1017] = 1043.07; $p$ < .0001. CFI = .55; GFI = .88; NFI = .53; RMSEA = .12 [90% CI = .111-.123]). Removal of non-significant pathways and addition of social media resulted in the model achieving a parsimonious goodness of fit ($\chi^2$ [19, 1017] = 77.58; $p$ < .0001. CFI = .96; GFI = .98; NFI = .95; RMSEA = .055 [90% CI = .043-.068]). Invariance tests revealed that the newly revised model differed across gender, age, level, competition status, and length of sport participation.

### Conclusion

This study showed that the formation of disordered eating symptomology might not be associated with sport pressures experienced by athletes. It revealed that disordered eating

**Competing interests:** The authors have declared that no competing interests exist.

development varies across gender, competition level, sport type, and age, which must be considered to prevent and treat disordered eating in athletes.

## Introduction

Participation in competitive sports has the potential to increase the risk of eating disorders and disordered eating in athletes [1, 2]. Disordered eating and eating disorders affect the psychological and physical health of millions worldwide [3]. Understanding, this particular pathology in athletes is especially intriguing as regimented diet and intense exercise is often part of competitive sport, but can also be symptoms or maintenance factors of disordered eating and eating disorders [4]. In 2007 (and later re-released in a second edition in 2012), Petrie and Greenleaf published a theoretical etiological model that outlined potential risk factors for the development of disordered eating in athletes as a series of mediators and moderators (see Fig 1). This model was chosen for additional study in this research due to its detailed nature and because it is the only disordered eating model that has been created specifically for athletes rather than a clinical model that has been adapted for use with athletes [5]. Furthermore, as dictated by the original authors of the theoretical etiological model, more testing in athlete samples is needed. This theoretical etiological model is based on the dual-pathway model that posits that social pressure and the internalisation to be thin lead to body dissatisfaction, which in turn leads to negative affect and dietary restraint resulting in eating pathology [6, 7]. The theoretical etiological model also uses evidence from research that explored psychological and social risk factors for disordered eating in athlete and non-athlete samples that consisted primarily of female participants [7–10]. The theoretical etiological model outlines eight factors/ mediators that are deemed to be risk factors or causal risk factors based on previous experimental or longitudinal research in the topic area [6]. The factors that are included in the model are (1) sport pressures, (2) societal pressures, (3) internalisation, (4) body dissatisfaction, (5) negative affect, (6) restrained eating, (7) modelled behaviours by peers and family, and (8) binge eating and bulimia. The model also includes five groupings of moderators that affect the intensity and directionality between factors, however this research will focus on the eight risk factors outlined in detail below.

### Sport pressures

In the original chapter that accompanies the theoretical etiological model, the higher prevalence of disordered eating in athletes was determined to result from three elements: being an athlete versus a non-athlete, competition level, and type of sport (lean/non-lean)[11]. These three elements that influence prevalence are used in the current study to operationalise sport pressure alongside three additional relevant elements: the number of years of sport participation, training hours per week, and whether an athlete is currently competing or in off-season. These are applied to help to determine whether participating for longer in sport and actively competing creates additional pressure that may relate to the development of disordered eating behaviour.

 Previous research findings have been largely consistent in indicating that for lean sports, in which performance or success is influenced by a lean body shape, such as gymnastics or diving, disordered eating has a higher prevalence rate for both men and women (e.g., [12–18]). The level at which the athlete competes is also a relevant sport pressure. However, while some

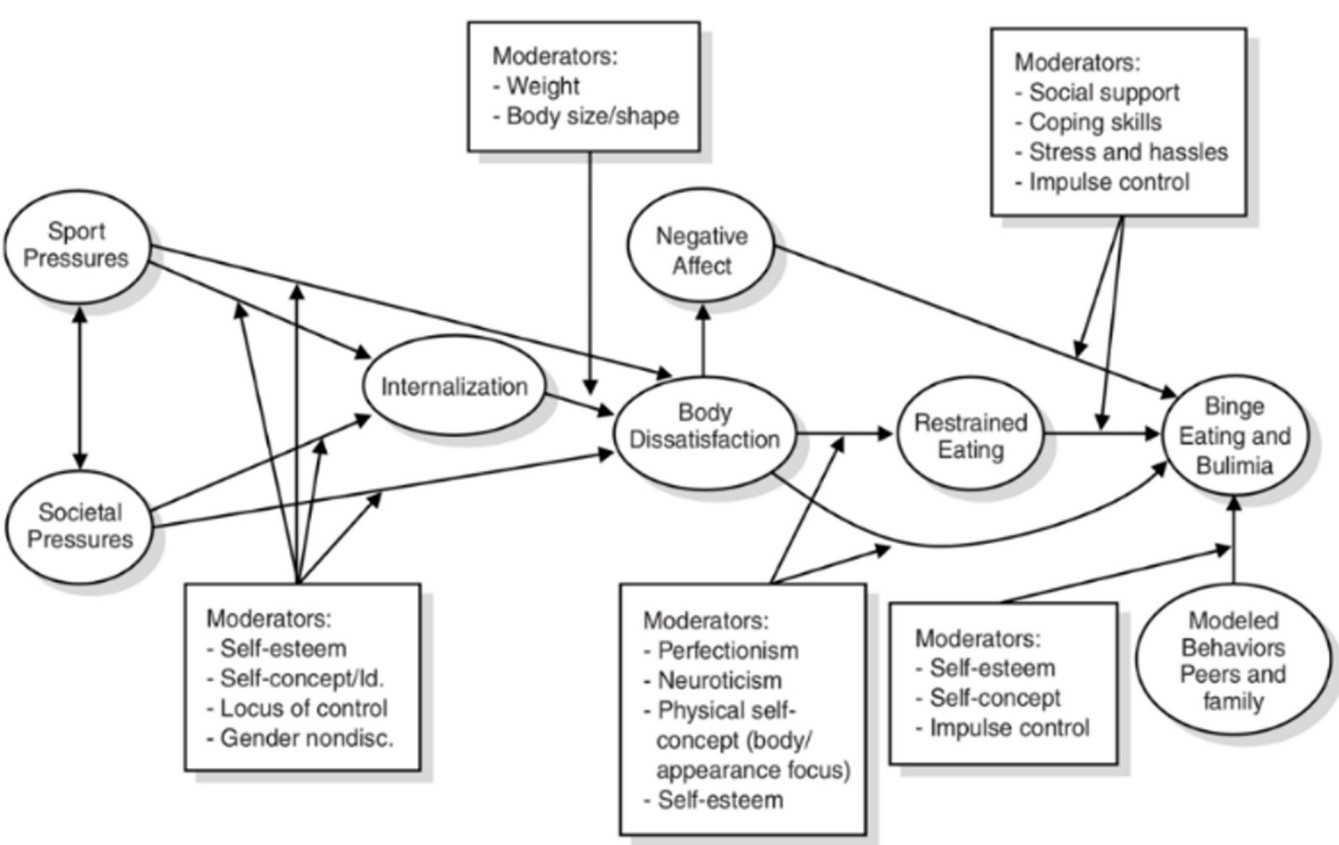

**Fig 1. Original theoretical etiological model from petrie and greenleaf, 2007, 2012.**

research has posited that elite athletes have higher levels of disordered eating than recreational ones, other research has found the opposite [2, 12, 19].

The current study also includes the number of hours spent training per week as a sport pressure, which has rarely been included in previous research, as much of the previous work has been conducted using National Collegiate Athletic Association athletes in the U.S. (whose hours of training are capped at 20 per week) [20]. This study is relatively unique in its investigation of the amount training hours per week as a sport pressure and will build on previous work which claims that level and training regime should be investigated [21]. The original work by Petrie and Greenleaf stated that increased exercise and training is inversely connected with body dissatisfaction and is thus a relevant factor to be included [22]. Therefore, increased hours could be assumed to mean an increased amount of pressure on the athlete with the caveat that not all sports require more than 20 hours of training per week. Furthermore, there is a positive correlation between level and hours per week, with more hours generally required for high-level sports; hence, these two variables tend to be related. The present study also determines whether an athlete's status as currently competing or currently in 'off-season' plays a role as a sport pressure. Each sport has a time of year or season designated for competition. For example, rugby players compete in the winter, while major cycling events are held during the summer. Although athletes continue to train during the off-season, it has been suggested that measuring disordered eating without accounting for whether those athletes are currently feeling the pressures of competition may have resulted in equivocal findings [23, 24].

## Societal pressures

Societal pressures affect athletes, and much like the rest of the population in the Western world, female athletes feel pressure to fit a thin ideal, and men feel pressure to fit a muscular one [25]. However, for athletes, these ideals can compound the sport pressures already exacted upon them, or an opposing tension can arise [26–28]. For example, for a female powerlifter, gaining mass may be advantageous for her sport but not for meeting societal ideals. Societal pressures can stem from family, teammates, the media, and more recently, from social media and the internet [29, 30, 21, 31]. Discerning the degree to which these pressures are internalised—the extent to which the pressures are incorporated into one's values and beliefs—is key to understanding the impact of sport and societal pressures on athletes. It is often only when these pressures are internalised that athletes' body satisfaction is damaged and disordered eating behaviours become likely to develop [32, 33].

The current study updates the model to fit the demands of the 21st century by including social media pressure as one of the societal pressures faced by athletes. The majority of popular social media involves photographs, and research has shown that viewing photographs decreases self-evaluations in women and is linked to the increased prevalence of eating disorder symptoms, including excessive exercise [33–37]. Social media creates a community in which harmful behaviours associated with disordered eating and eating disorders can be reinforced or seen as commonplace [35]. In fact, those people considered to be part of the online health community—a source of 'fitspiration'—scored higher on disordered eating questionnaires and presented more symptoms of compulsive exercise than those whose online presence was focused on travel [38]. Research indicates that repeated exposure to photographs of women who possess the athletic-ideal body type increases body dissatisfaction, and as athletes are likely to follow other athletes on social media, the implications of social media usage warrant attention [39].

## Internalisation and body dissatisfaction

Internalisation, in this context, is the incorporation of an external, often unattainable body shape ideal into how one measures one's own self-worth [11]. When sport and societal pressures give rise to an ideal body standard that is then internalised, if an athlete perceives his or her own body to not match to this ideal, body dissatisfaction and disordered eating behaviours can occur [40–42]. This body dissatisfaction has been found to be directly related to the formation and maintenance of eating disorders and disordered eating [8]. Evidence specifically suggests that body dissatisfaction in athletes is strongly related to the development of disordered eating [14, 43]. However, discrepancies regarding which type(s) of athletes experience the most body dissatisfaction and whether athletes experience more dissatisfaction than the general population remains the subject of debate [17, 44–46]. When comparing levels of body dissatisfaction in athletes to nonathlete controls and lean sport athletes to nonlean sport athletes only equivocal conclusions can be drawn; however, despite the debate, there is consensus that body dissatisfaction contributes to disordered eating [14, 17].

## Negative affect

While negative affect is presented in the original model as a link between body dissatisfaction and bulimic behaviours based on research in the general population, few studies have examined negative affect in athletes in relation to disordered eating [11, 47]. Research has found that negative affect in athletes, especially constructs such as fear and guilt, influenced bulimic behaviour when tested in conjunction with increased body dissatisfaction and dietary intent [48–50]. Negative affect has been linked with disordered eating in athletes via increased levels

of compulsive exercise [51, 52]. As athletes have a readily available outlet of sport and exercise, over-exercise may be a tempting avenue through which athletes alleviate negative affect [4].

## Modelled behaviours

Modelled behaviours are those adopted behaviours that have been reinforced as the norm in a group and those modelled behaviours pertaining to eating psychopathology have been shown to increase disordered eating symptomology [53, 54]. The mediator that is termed modelled behaviour by peers and family is one that much of the literature has overlooked, and research has yet to operationalise or validate a measurement tool for this potential risk factor. Therefore, the present research begins to close this gap in the literature by including questions relating to modelled behaviours of disordered eating among athletes.

## Restrained eating, and binge eating and bulimia

Restrained eating refers to the attempted and successful behaviour of limiting food intake in terms of quantity and type [55]. Binge eating and bulimia refer to the symptoms of overeating and compensatory behaviours [56]. These two mediators are conceptually considered part of disordered eating but are examined separately as dictated by the theoretical model. In the model, restrained eating is thought to mediate the relationship between negative affect and binge eating and bulimia [11, 47]. Restrained eating, as well as bulimia and bingeing, have been linked to negative affect resulting from stressful situations [57].

## Previous structural testing of the model

Previous studies have tested several elements of the model [58]. However, only a few have used structural equation modelling that allows for the simultaneous analysis of relationships, which is required to move the field forward. First, Anderson, Petrie, and Neumann (2011) tested the eight factors depicted in the model in female collegiate gymnasts, swimmers, and divers. They found that modelled behaviours did not fit the model and that several pathways required adjustment in that sport pressure directly impacted body dissatisfaction and dietary restraint rather than being mediated by internalisation. Several years later, De Sousa Fortes and colleagues (2015) considered the model in a multi-sport sample of male adolescent Brazilian athletes. The researchers originally predicted that the sport pressures of training regimes at the competitive level, body fat percentage, and sociocultural pressures would lead to body dissatisfaction, which in turn may promote disordered eating in male athletes. However, they found that only the sociocultural factors and body dissatisfaction predicted disordered eating. These two studies, the prevalence factors taken from the original 2007 chapter, and the findings of a literature review were all used to guide the operationalisation in the present study to create measurement consistency throughout the research area [2, 59].

## Demographic correlates of disordered eating

The current study also aims to test this model between gender and age. Previous studies have tested the elements of the model in single-gender samples [21, 48, 60, 61]. Therefore, in an effort to advance the field this study has included both genders for analysis. The literature on disordered eating has shown higher prevalence rates of disordered eating and eating disorders in female athletes, but a co-ed sample is needed to re-test this element [2]. Due to convenience sampling, the ages of the athletes in previously conducted studies have often been of traditional US university age (*c.f.* [28, 62]). This study instead considers a broader range of ages to ascertain how age may influence disordered eating development. With regard to this demographic

information, elements that are intrinsic to each individual will help to enhance how this research can be utilised in applied care.

## Aims

The existing literature on athletes and disordered eating has presented significant inconsistencies with respect to the utilised tools, methodologies, and general findings. There is also a lack of longitudinal studies in the literature. As a result, it is difficult to identify the underlying causal factors for disordered eating in this population [63, 59]. The original theoretical etiological model is comprehensive in its inclusion of factors and its sport-specific focus, thus providing an ideal starting point. In summary, this study has three aims. The first is to use structural equation modelling (SEM) to determine the utility of the original theoretical etiological model by testing all eight mediators in a diverse sample that includes both men and women and a wide range of sports and levels of participation. The second aim is to develop and test a revised model that includes social media. The third and final aim is to test the newly revised model and determine whether it is equivalent across groups such as gender, sport type, competition level, if currently competing, years of participation, and age.

## Methods

### Procedure

This study received ethical approval from the Clinical Educational and Health Psychology Department at University College London, Reference for this approval: CEHP/2018/573. Participants were recruited using online social media campaigns as well as by word-of-mouth. Those who identified themselves as athletes and performed at least ten hours of training per week were invited to participate. The questionnaire consisted of 241 questions and took just over 30 minutes to complete. It was administered using Opinio [64] and following informed consent, participants could start the questionnaire and return to it at a later point within a seven-day window. The questionnaire was open from January 27, 2019 to February 24, 2019. Participants who completed the questionnaire were given a £5 Amazon voucher as compensation for their time.

### Participants

The inclusion criteria that were applied for this study were that participants had to be over 18; there was no upper cap on the age of potential participants. Participants also had to consider themselves to be an athlete, and a minimum of ten hours a week participating in their sport as well as have been actively competing, thus meaning that competitive sport—rather than zealous exercise—was a significant part of their daily experience. After providing their informed consent, 1,208 participants started the online questionnaire, of whom 1,084 completed it. Only completed questionnaires were kept for analysis. Seventeen participants were excluded due to not fitting the inclusion criteria and another three were deleted as they were based outside of the UK, as this study did not have ethical approval to recruit internationally. Thirty-five more with matching email addresses and exact matching answers were removed due to suspicion from the researcher that the answers were duplicates completed to obtain the £5 gift voucher. A final two were removed as when asked, they did not identify as an athlete. Therefore, the final sample consisted of N = 1017 athletes, of whom 56% were male and 44% were female. A wide range of sports was represented in the 1,017 participants, with swimming (18%), tennis (8%), football (soccer) (11%), basketball (18%), volleyball (10%), dancing (7%), and various athletics (track and field) events (19%) making up the majority of the sports represented.

Other less-represented sports included golf, boxing, equestrian, cycling, ping-pong, orienteering, rugby, rowing, and race walking. Slightly over half (58.3%) of the participants were classified as competing in lean sports, with 41.7% classified as non-lean sport participants. Several other demographic variables as well as clinical scores were captured and are presented in Table 1.

## Measures

The materials used to operationalise the relevant variables were a combination of existing validated questionnaires and other new measures created for the purpose of this study. It is important to note that the reliability and validity of these measures were largely established in non-athlete samples.

## Demographics

Participants' age, gender, sport type information, years spent competing, the teams for which they had competed, and self-reported height and weight were collected.

## Sport pressure

Sport pressure was operationalised as lean (where weight and shape are integral to performance success) or non-lean sport participation, the level at which the participants competed, the hours per week that they spent training, their years of participation in that sport, and their current status as competing or in off-season. The categorisation of lean and non-lean sports was decided by the first author, based on previous classifications of sports in the existing literature or based on the first author's applied experience. The hours trained per week were categorised into 10–15, 16–25, 26–40, and 40+ hours. The levels of sport participation were grouped

**Table 1. Additional demographic information.**

|  |  | Percent of Total |
|---|---|---|
| Age | 18–26 | 85.7% |
|  | 27+ | 14.3% |
| Individual or Team | Individual | 52.4% |
|  | Team | 47.6% |
| When in Season | Currently Competing | 75.2% |
|  | Not Currently Competing | 24.8% |
| Years participated in their sport | 1–3 Years | 6.4% |
|  | 4–8 Years | 52.7% |
|  | 9–15 Years | 40.1% |
|  | 16+ Years | 0.8% |
| Hours/Week | 10–15 hours/week | 17.9% |
|  | 16–25 hours/week | 60.4% |
|  | 26–40 hours/week | 20.9% |
|  | 40+ hours/week | 0.8% |
| Level | NonElite | 82.5% |
|  | Elite | 17.5% |
| BMI (Range) | All Participants | 14.1–32.4 |
| BMI (Mean; SD) | All Participants | 21.2; 2.5 |
| EDE-Q (Range) | All Participants | 0.39–5.49 |
| EDE-Q (Mean; SD) | All participants | 2.6; 0.8 |

into 'non-elite', which was comprised athletes competing at a county or regional level, and 'elite', which consisted of those participating at a national or international standard. Athletes determined whether they were part of an individual or team sport, and in the instances where the athletes chose 'both', the authors used their discretion to categorise that response appropriately. Athletes could choose from the following categories of the years for which they had taken part in their sports: 1–3 years, 4–8 years, 9–15 years, and 16+ years. Finally, athletes could indicate whether they were currently competing or in their off-season. In the case that N/A was chosen for the answer, it was assumed that those athletes were consistently competing throughout the year and thus were categorised with those who indicated that they were currently competing.

### Societal pressure

Societal pressure was measured using the nine-item Information and seven-item Pressures subscales from the Sociocultural Attitudes Towards Appearance Questionnaire-3 (SATAQ-3) [65]. The subscales were summed for a total societal pressures score. An example of a statement from the pressure subscale is "I've felt pressure from TV or magazines to lose weight", and from the information subscale, "Pictures in magazines are an important source of information about fashion and 'being attractive.'" Cronbach alphas were high for these subscales as well as for the entire SATAQ scale: Information ($\alpha$ = .94), Pressures ($\alpha$ = .94), and global score ($\alpha$ = .94) [66]. Statements related to social media usage were also utilised with similar questions to those that were asked in the SATAQ about TV or magazines, simply by replacing these words with "social media". For example, "I've felt pressure from social media to be thin." This brief part of the questionnaire consisted of five items, was created for this study, and has not been formally validated; however, reliability calculations for the current study showed an alpha of .75.

### Internalisation

Internalisation was also measured by using subscales of the SATAQ-3, specifically the five-item Internalisation-Athlete ($\alpha$ = .89) and the nine-item Internalisation-General ($\alpha$ = .92) [66]. As above, these two subscales were summed for ease of analysis. An example item from the Internalisation-General is "I compare my body to the bodies of people who are on TV," and from the athlete scale, "I wish I looked as athletic as sports stars."

### Body dissatisfaction

The nine-item body dissatisfaction subscale (EDI-BD) of the Eating Disorder Inventory (EDI-2) was utilised to measure body dissatisfaction ($\alpha$ = .88) [67, 68]. An example of a statement from this subscale is "I think that my stomach is too big." Notably, this scale shows lower alpha coefficients for men compared to women [69].

### Negative affect

Negative affect was measured using the 10-item subscale ($\alpha$ = .85; 95%, CI = .84–.87) from the Positive and Negative Affect Schedule (PANAS), in which participants had to rate the extent to which they felt various emotions, such as "guilty", "scared", and "nervous: from "*Very slightly or not at all*" (1) to "*Extremely*" (5). Alpha reliability for this scale was .87.

### Restrained eating

Restrained eating was operationalised by using the Restraint Subscale of the Eating Disorder Examination Questionnaire (EDE-Q-R) [70, 71]. This subscale had high internal consistency in this sample ($\alpha$ = .78) and a high test-rest reliability in other mixed-gender samples ($r$ = .81) [72]. An example question from this subscale is "Have you **tried** to exclude from your diet any foods that you like in order to influence your shape or weight (whether or not you have succeeded)?." It is important to note that one question from this subscale was omitted due to human error, hence, as recommended by the authors of the scale, mean calculations were used with items included. The item missing was "Have you wanted your stomach to be empty?". It is also key to note that controversy over the content validity of this scale exists, with research indicating that it more accurately measures restriction rather than restraint [55, 73]. The global EDE-Q score was also calculated for each participant to better understand prevalence rates and create clinical relevance ($\alpha$ = .91).

### Modelled behaviours

No previous research has included a validated measure for modelled behaviours of peers and family, so this current study used seven questions to attempt to capture these sentiments. Example statements included "My friends diet or use weight control behaviours" and "My teammates diet or use weight control behaviours." Reliability analysis for the current study found the alpha to be .61. In the current study, questions were designed based on the qualitative results that had previously discussed the harmful role that modelled behaviours of teammates can have in the development of eating disorders [54].

### Binge eating and bulimia

Binge eating and bulimia were measured by using the bulimia subscale of the 64-item EDI-2 (EDI-B) [67]. This subscale is scored from "Never" to "Always" on a six-item Likert scale. An example statement is "I have gone on eating binges where I have felt that I could not stop." This subscale also has a high internal consistency ($\alpha$ = .76).

### Data analysis

SEM was conducted using AMOS. Several indicators of model fit were utilised during the analysis: $\chi^2$ significance, the Comparative Fit Index (CFI), Goodness of Fit (GFI), Normed Fit Index (NFI), and Root Mean Square Error Approximation (RMSEA). In general, a nonsignificant $\chi^2$ value shows that the data fits the model well, however, with large samples it is unlikely to obtain a p value < .05 [74]. For CFI, GFI, and NFI a value >.90 is considered to indicate acceptable fit and for RMSEA a value < .08 is needed for acceptable fit [75].

## Results

Analysis was conducted using SPSS version 25 and the SPSS add-on AMOS. Excel Macros was also utilised. When analysing the data for normality, it was determined that all variables had distributions within the acceptable range for skewness and kurtosis [75].

### Descriptive statistics

Table 2 outlines the descriptive statistics and intercorrelations for all the variables involved in the model. While some of the correlations did show significant relationships between variables the correlations were all weak, suggesting that variables were measuring relatively independent underlying constructs.

**Table 2. Table of correlations and means and standard deviations.**

| | Lean/NonLean | Hours/Week | Elite/Nonelite | Competing/Offseason | Years done sport | SATAQ-Pressures | SATAQ-Information | Social Media Pressures | Internalisation-General | Internalisation-Athlete | EDI-BD | Modelled Behaviour | EDE-Q Restraint Sore | EDI-B |
|---|---|---|---|---|---|---|---|---|---|---|---|---|---|---|
| Lean/NonLean | 1 | .183** | 0.051 | .069* | .096** | −0.024 | 0.061 | 0.001 | 0.028 | 0.033 | −0.013 | .073* | 0.048 | 0.014 |
| Hours/Week | | 1 | .256** | −.321** | .296** | −0.003 | −0.002 | 0.014 | 0.052 | −.072* | 0.015 | −.154** | −.209** | .107** |
| Elite/Nonelite | | | 1 | −0.013 | .163** | 0.023 | .084** | −0.032 | .090** | 0.027 | −0.049 | −0.005 | −.087** | .097** |
| Competing/Offseason | | | | 1 | −.191** | −0.016 | 0.038 | −0.050 | −0.051 | −0.005 | −0.031 | .275** | .282** | −0.025 |
| Years done sport | | | | | 1 | −.113** | −0.049 | −0.057 | 0.027 | −.119** | .069* | −.077* | −0.061 | 0.013 |
| SATAQ-Pressures | | | | | | 1 | .372** | .530** | .479** | .454** | −0.014 | −0.016 | −0.020 | .395** |
| SATAQ-Information | | | | | | | 1 | .310** | .337** | .193** | −.069* | 0.008 | 0.009 | .207** |
| Social Media Pressures | | | | | | | | 1 | .416** | .423** | −0.005 | −0.008 | 0.005 | .276** |
| Internalisation-General | | | | | | | | | 1 | .312** | −0.001 | −0.013 | −0.028 | .230** |
| Internalisation-Athlete | | | | | | | | | | 1 | −0.016 | 0.001 | −0.030 | .303** |
| EDI-BD | | | | | | | | | | | 1 | −0.028 | 0.017 | 0.051 |
| Modelled Behaviour | | | | | | | | | | | | 1 | .564** | −0.022 |
| EDE-Q Restraint | | | | | | | | | | | | | 1 | −0.015 |
| EDI-B | | | | | | | | | | | | | | 1 |
| M | | | | | | 22.33 | 28.04 | 17.67 | 28.18 | 16.15 | 6.26 | 15.03 | 3.03 | 4.33 |
| SD | | | | | | 3.33 | 3.07 | 3.19 | 3.14 | 2.35 | 2.72 | 3.27 | .94 | 3.57 |

\* p < .05;

\*\* p < .001

## Clinical information

The gold standard for clinically diagnosing an eating disorder is by using the EDE-Q global along with the accompanying Eating Disorder Examination Interview [76]. Therefore, this study cannot clinically diagnose as it only reports the EDE-Q global, however it can give clues to prevalence rates among athletes with 5.7% of the current sample scoring above the suggested clinical cut off of 4.0 [77, 78]. Additional information about the EDE-Q global scores can be found in Table 1.

## Testing the theoretical etiological model

Testing the theoretical etiological model involved creating a hybrid model, a cross between a path model and a measurement model in AMOS (Fig 2). Results revealed that this model did not fit the data well. $\chi^2$ (70, 1017) = 1043.07; $p < .0001$. CFI = .55; GFI = .88; NFI = .53; RMSEA = .12 (90% CI = .111--.123). Regression weights are indicated with Beta's to show the strength of the relationship between factors.

To enhance the model, social media was added as an observed variable as part of social pressures. This did improve the fit of the model but not to an acceptable standard. $\chi^2$ (70, 1017) = 646.09; $p < .0001$. CFI = .74; GFI = .92; NFI = .72; RMSEA = .09 (90% CI = .084--.096). Thus the original etiological model with social media was rerun followed by the deletion of all nonsignificant pathways, followed by other pathway alternations as indicated by the modification indices. This resulted in the following revised model with excellent parsimonious goodness of fit (Fig 3). This was the model used for invariance testing. $\chi^2$ (19, 1017) = 77.58; $p < .0001$. CFI = .96; GFI = .98; NFI = .95; RMSEA = .055 (90% CI = .043--.068).

Standardised regression weights, or direct effects are shown for this newly revised model in the figure (Fig 3). Indirect effect analysis along with bootstrapping 200 samples at a 90% confidence interval revealed that Societal Pressures ($\beta = .45$, $p = .002$) and Body Dissatisfaction ($\beta = .08$, $p = .01$) both had significant indirect effects on Bulimia.

## Testing for invariance across groups

The revised model created (Fig 3) had the best fit of all the models tested. Therefore, tests of invariance across several groups including gender, age, competition level, if currently competing, and years done sport, using this revised model are outlined below. As sport pressure had been removed from the model due to a lack of fit, the model could be tested for differences

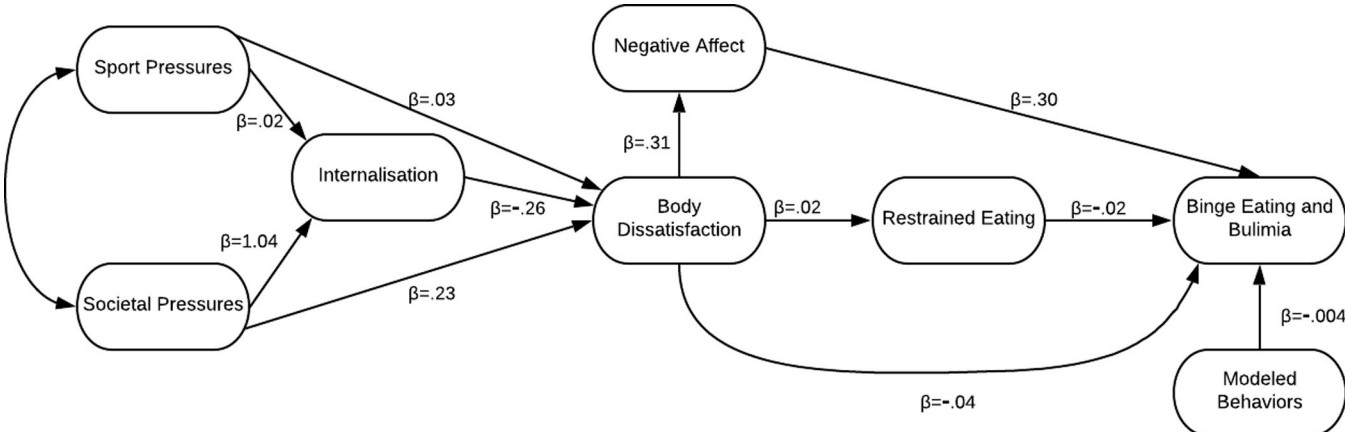

**Fig 2. Original theoretical etiological model in athletes.**

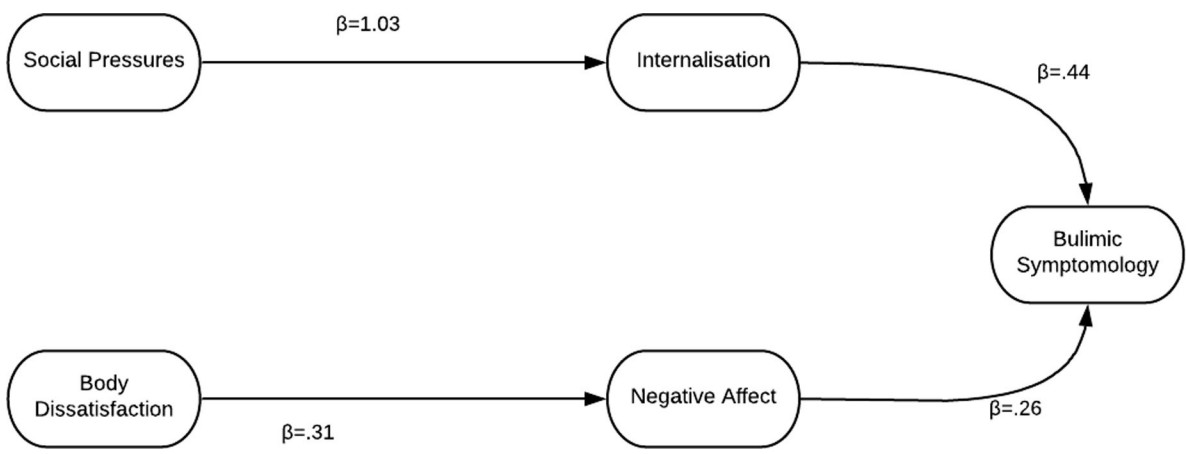

**Fig 3. Revised model with best fit for athletes, used for all invariance testing.**

across these pressures as well. When testing for invariance, two methods were employed. First, AMOS multigroup analysis was utilised to examine the Comparative Fit Index to see if the constrained models differed from the unconstrained model by an amount equal or larger than .01, which is then taken as indication that the model differs across groups [74, 79]. Secondly, a Chi Square difference test using AMOS analysis and Excel Macros was also used to give additional insight. While still showing good fit for the vast majority of groups, the model showed that variance was present depending on gender, age, level of athlete, whether athlete was currently competing, and how long that athlete had participated in sport. The model was invariant for lean and nonlean sport types.

Multigroup invariance testing revealed differences between genders (see Table 3). The model fit better for men $\chi^2$ (19,567) = 54.46; $p < .0001$; CFI = .96; GFI = .98; NFI = .95; RMSEA = .06 (90% CI = .040–.076) than for women $\chi^2$ (19, 450) = 46.70, $p< = .0001$; CFI = .95; GFI = .98; NFI = .93; RMSEA = .06 (90% CI = .037–.078). Excel macros used in conjunction with AMOS revealed that gender differences were explained by differences in the pathway between body dissatisfaction and negative affect at a 99% confidence interval (CI).

Invariant testing between those athletes under age 26 (young) and those above age 26 (mature) also showed a significantly different fit (see Table 4). The model fit better for younger athletes $\chi^2$(19, 872) = 79.81, $p < .0001$; CFI = .96, GFI = .98, NFI = .94, RMSEA = .061 (90% CI = .047–.075) than mature ones $\chi^2$(19, 145) = 36.98, $p = 0.008$; CFI = .93, GFI = .94, NFI = .87, RMSEA = .081 (90% CI = .041–.120). Further testing using excel macros revealed that age specifically moderated the pathways between internalisation and bulimia and negative affect and bulimia at a 99% CI thus explaining the different fit for the two age groups.

**Table 3. Gender model fit indices across various model constraints.**

| Model | $\chi^2$ | DF | CFI | GFI | NFI | RMSEA | CI for RMSEA | |
|---|---|---|---|---|---|---|---|---|
| Unconstrained | 101.170 | 38 | .960 | .977 | .938 | .040 | .031 | .050 |
| Measurement weights | 107.601 | 42 | .959 | .975 | .935 | .039 | .030 | .048 |
| Structural weights | 130.870 | 45 | .946 | .969 | .920 | .043 | .035 | .052 |
| Structural covariances | 131.040 | 46 | .946 | .969 | .920 | .043 | .034 | .051 |
| Structural residuals | 132.603 | 49 | .947 | .969 | .919 | .041 | .033 | .049 |
| Measurement residuals | 149.089 | 55 | .941 | .965 | .909 | .041 | .033 | .049 |

**Table 4. Age model fit indices across various model constraints.**

| Model | $\chi^2$ | DF | CFI | GFI | NFI | RMSEA | CI for RMSEA | |
|---|---|---|---|---|---|---|---|---|
| Unconstrained | 116.904 | 38 | .951 | .973 | .930 | .045 | .036 | .055 |
| Measurement weights | 138.229 | 42 | .940 | .968 | .917 | .048 | .039 | .056 |
| Structural weights | 160.925 | 45 | .928 | .963 | .904 | .050 | .042 | .059 |
| Structural covariances | 163.515 | 46 | .927 | .964 | .902 | .050 | .042 | .059 |
| Structural residuals | 167.979 | 49 | .926 | .962 | .899 | .049 | .041 | .057 |
| Measurement residuals | 178.351 | 55 | .924 | .961 | .893 | .047 | .039 | .055 |

The model better explained disordered eating for nonelite athletes $\chi^2(19, 839) = 78.56$, $p < .0001$; CFI = .95, GFI = .98, NFI = .94, RMSEA = .061. (90% CI = .047−−.076) than elite athletes $\chi^2(19, 178) = 57.68$, $p < .0001$; CFI = .88, GFI = .93, NFI = .84, RMSEA = .107 (90% CI = .076−−.139) with elite athletes being the only group in which the CFI dipped below the level required for adequate fit. See Table 5 for invariant test results. Excel macros analysis showed that for these groups, the difference could be explained by the path from societal pressures to internalisation at a 90% CI and from internalisation to bulimia at a 95% CI.

Those currently competing and those in off-season were variant in a multi-group analysis test (see Table 6) but only showed a very slight difference in fit for the GFI, but no difference in fit for the CFI with those currently competing showing a fit of $\chi^2(19, 252) = 41.40$, $p = .002$; CFI = .95, GFI = .96, NFI = .91, RMSEA = .069 (90% CI = .040−−.097) and those in offseason with a fit of $\chi^2(19, 765) = 73.02$, $p < .0001$; CFI = .95, GFI = .98, NFI = .94, RMSEA = .061 (90% CI = .047−−.076). Excel macros showed that the difference in fit can be explained by currently competing or being in offseason moderating the pathways from internalisation to bulimia and negative affect to bulimia at a 95% CI and body dissatisfaction to negative affect at a 99% CI.

Finally, the model fit increasingly well the longer an athlete had participated in his/her sport (see Table 7). Those doing sport for one to three years showed a fit of $\chi^2(19, 65) = 33.47$, p = .02; CFI = .84, GFI = .89, NFI = .72, RMSEA = .109. (90% CI = .042−.169), those doing it for four to eight years showed a fit of $\chi^2(19, 536) = 106.22$, p < .0001; CFI = .91, GFI = .96, NFI = .89, RMSEA = .093 (90% CI = .076-.110), and those who had done sport for nine or more years $\chi^2(19, 416) = 76.75$, p < .0001; CFI = .92, GFI = .96, NFI = .90, RMSEA = .086 (90% CI = .066−−.106). (An insufficient number of athletes had done their sport for over 16 years and so the 16 + category was combined with the 9–15 years category for analysis). Excel macros exposed that years participated in sport influenced the pathways from internalisation to bulimia at a 99% CI and negative affect to bulimia at a 95% CI and body dissatisfaction to negative affect at a 90% CI.

## Discussion

The purpose of this research was to assess the applicability of the theoretical etiological model proposed by Petrie and Greenleaf in 2007 and 2012 in a large, multi-sport, mixed-gender

**Table 5. Elite vs Nonelite model fit indices across various model constraints.**

| Model | $\chi^2$ | DF | CFI | GFI | NFI | RMSEA | CI for RMSEA | |
|---|---|---|---|---|---|---|---|---|
| Unconstrained | 136.387 | 38 | .939 | .969 | .919 | .051 | .042 | .060 |
| Measurement weights | 155.302 | 42 | .930 | .964 | .908 | .052 | .043 | .060 |
| Structural weights | 156.395 | 45 | .931 | .963 | .907 | .049 | .041 | .058 |
| Structural covariances | 160.276 | 46 | .930 | .963 | .905 | .049 | .041 | .058 |
| Structural residuals | 164.555 | 49 | .929 | .963 | .902 | .048 | .040 | .056 |
| Measurement residuals | 208.583 | 55 | .906 | .958 | .876 | .052 | .045 | .060 |

**Table 6. Currently competing vs out of season model fit indices across model constraints.**

| | $\chi^2$ | DF | CFI | GFI | NFI | RMSEA | CI for RMSEA | |
|---|---|---|---|---|---|---|---|---|
| Unconstrained | 114.454 | 38 | .952 | .974 | .931 | .035 | .036 | .054 |
| Measurement weights | 121.960 | 42 | .950 | .972 | .926 | .034 | .035 | .052 |
| Structural weights | 139.598 | 45 | .941 | .969 | .916 | .037 | .038 | .054 |
| Structural covariances | 141.597 | 46 | .940 | .968 | .915 | .037 | .037 | .054 |
| Structural residuals | 145.959 | 49 | .939 | .967 | .912 | .036 | .036 | .053 |
| Measurement residuals | 164.633 | 55 | .932 | .964 | .901 | .037 | .037 | .052 |

sample. This study specifically considered relationships between the eight mediators described in the theoretical model: sport pressure, societal pressure, internalisation, body dissatisfaction, negative affect, modelled behaviours, restrained eating, and binge eating and bulimia, as well as the addition of social media as a societal pressure. This study aimed to expand upon previous research that tested this theoretical model by adding social media in order to create a newly revised parsimonious model with high goodness of fit. It was hoped that by using a larger sample of mixed gender athletes that was not constrained by the convenience sampling of US universities would mean a revised, better fitting model would be revealed. The revised model's utility was also assessed across groups to test for invariance.

The analysis showed poor fit for the original theoretical etiological model, although this fit improved slightly with the inclusion of social media. The relevance of social media as a source of societal pressure has been shown in a range of recent research that has indicated how intimately sociocultural experiences are tied to the online world [80]. The next step was to remove non-significant pathways, which resulted in the creation of a revised model that had good fit on the CFI, GFI, and other relevant indices.

This newly revised model showed that societal pressures (including social media), mediated by internalisation, are associated with binge eating and bulimia, while body dissatisfaction leads to bulimia and binge eating, mediated by negative affect. It is important to note that restrained eating was eliminated from the model and, therefore, this model predicts binge eating and bulimia as only one facet of disordered eating in athletes. Predicting only bulimia symptomology matches with the findings of the dual-pathway model that was the original clinical basis for all modelling by Petrie and Greenleaf [57].

As this study is cross-sectional, causal claims cannot be made. However, these results demonstrate that the formation of binge eating and bulimia is not associated with the sport pressures experienced by athletes, as theoretically suggested. These findings add to the growing body of research that indicates that sport pressures may not directly influence the development of disordered eating symptoms of binge eating and bulimia [5, 21]. The applied implication of this specific finding may be that prevention techniques and interventions utilised in non-athlete samples may be applicable. However, it is still important to note that several aspects of

**Table 7. 'Years spent participating in sport' model fit indices across model constraints.**

| Model | $\chi^2$ | DF | CFI | NFI | GFI | RMSEA | CI for RMSEA | |
|---|---|---|---|---|---|---|---|---|
| Unconstrained | 216.700 | 57 | .910 | .883 | .952 | .053 | .045 | .060 |
| Measurement weights | 357.694 | 65 | .835 | .808 | .922 | .067 | .060 | .073 |
| Structural weights | 403.791 | 71 | .813 | .783 | .909 | .068 | .062 | .075 |
| Structural covariances | 410.205 | 73 | .810 | .779 | .910 | .067 | .061 | .074 |
| Structural residuals | 461.764 | 79 | .784 | .752 | .903 | .069 | .063 | .075 |
| Measurement residuals | 520.471 | 91 | .758 | .720 | .891 | .068 | .063 | .074 |

sport pressure created variance in the model, demonstrating that while sport pressure does not fit within the model, various sport pressures may still be tangentially relevant to the development of disordered eating symptoms of bulimia and binge eating in athletes.

Invariance testing across several groups showed that the newly created model differed across several groups, and the results showed in which pathways these differences arose. This variance is something that must be taken into account when considering the prevention and treatment of disordered eating in athletes. With sport pressures removed from the model, invariance testing was also conducted across the factors that were originally described as measuring sport pressure.

While the newly revised model showed very good fit for both males and females, tests of gender differences revealed that the model fit better for male than for female athletes, which is notable, as the original model was designed based on literature that included mainly females. However, research has found increasing evidence that significant numbers of male athletes suffer from disordered eating [1, 81]. The model also fit better for young athletes, specifically those younger than age 27, which is also noteworthy, as the vast majority of research on athletes originates from convenience sampling in US universities, where the ages are often limited to 18–22 years. The model also fit better for those athletes of a non-elite level (i.e., those at a regional or county level), which indicates that these athletes, who are serious yet aspiring, may not have adequate support in coping with the pressure that they experience, whereas elite athletes have access to such support. The model fit better for athletes who completed the questionnaire while currently in their off-season, something that is rarely taken into account when surveying athletes. This finding may indicate that when training intensifies around competition time, it provides an outlet for athletes that allows them to fully fuel their bodies without internalising societal pressures or experiencing negative feelings around food. This finding is in line with previous work that has found athletes score higher on DE measures in the pre-competition time of the season than when currently competing [82]. Finally, invariance testing showed that the longer an athlete had participated in a sport, the better the model fit for that athlete, indicating an increasing pressure on athletes who had participated in a sport for over nine years. Notably, the model was invariant across lean and non-lean sport types. This final lack of variance may be due to societal pressures playing a larger role in the creation of disordered eating symptomology rather than the sport classification as lean or nonlean. In other words, perhaps it is society's image of an athlete that creates pressure for weight and shape to be managed in a way that defies the physiological demands of the sport (Stoyel, Shanmuga-nathan-Felton, Stoyel, & Serpell, Under Review).

Invariance testing gives indications of how lived experiences in sport and therefore disordered eating symptoms can fluctuate sport to sport, level to level, and season to season. Noticing the variance between demographics of athletes acknowledges that formation, prevention, and treatment of disordered and eating disorders in this population is not one size fits all, and gives early clues as to how to best tailor care.

## Limitations

This study is limited by its cross-sectional nature; however, future longitudinal phases of this project on currently underway. The measurement scales used in the current study were the ones most commonly found in eating disorder and disordered eating literature; however, none were specifically designed for athletes. Perhaps this use of general scales can explain why restrained eating also no longer fit. Additionally, in the absence of other acceptable measures, the scales for social media usage and modelled behaviours were purpose-built for this study and thus were not formally validated in previous research and so had lower Cronbach alpha

coefficients than would normally be acceptable. However, the sample, the range of different sports, and the fact that this study did not rely on a convenience sample of US university athletes from a similar age-range make it a unique study in this area of research. Finally, while clinical information was collected and superficially analysed, this research is limited in its ability to make sound assumptions on clinical implications.

## Future directions

Future research should use a longitudinal design to determine whether the relevant factors are able to predict future disordered eating in an athlete sample, something that this research group aims to do over the course of the next year. Repeating this research with athletes younger than 18 years old would also provide additional insight as it is a high-risk population for the development of eating disorders. A qualitative investigation to further understand the differences between the original etiological model and the revised model will also be undertaken.

## Conclusion

In conclusion, this study provides a large amount of empirical data to further test the original theoretical model with a more diverse sample than previously used. In doing so, a newly revised model was created that can begin to explain the presentation of disordered symptomology specifically related to binge eating and bulimia in a wide range of male and female athletes. Future research should study samples in repeated measures designs so that it is possible to begin to predict the development of disordered eating symptoms in athletes.

## Supporting information

**S1 Table.**
(XLSX)

## Acknowledgments

The authors wish to acknowledge Dr Antony Cooper for his help and guidance.

## Author Contributions

**Formal analysis:** Hannah Stoyel.

**Supervision:** Lucy Serpell.

**Writing – original draft:** Hannah Stoyel.

**Writing – review & editing:** Hannah Stoyel, Vaithehy Shanmuganathan-Felton, Caroline Meyer, Lucy Serpell.

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
