## [Decision Letter · Decision Letter 0]

27 Feb 2020

PONE-D-19-34507

Psychological Risk Indicators of Disordered Eating in Athletes

PLOS ONE

Dear Hannah,

Thank you for submitting your manuscript to PLOS ONE. The manuscript was revised by two reviewers. They both recommended minor revisions and therefore I invite you to re-submit a revised version of this manuscript. 

We would appreciate receiving your revised manuscript by 26/03/2020. To enhance the reproducibility of your results, we recommend that if applicable you deposit your laboratory protocols in protocols.io, where a protocol can be assigned its own identifier (DOI) such that it can be cited independently in the future. For instructions see: http://journals.plos.org/plosone/s/submission-guidelines#loc-laboratory-protocols

We look forward to receiving your revised manuscript.

Kind regards,

Valentina

Valentina Cardi

Academic Editor

PLOS ONE

Journal Requirements:

Reviewers' comments:

Reviewer's Responses to Questions

**Comments to the Author**

1. Is the manuscript technically sound, and do the data support the conclusions?

Reviewer #1: Yes

Reviewer #2: Yes

2. Has the statistical analysis been performed appropriately and rigorously? 

Reviewer #1: Yes

Reviewer #2: Yes

3. Have the authors made all data underlying the findings in their manuscript fully available?

Reviewer #1: No

Reviewer #2: Yes

4. Is the manuscript presented in an intelligible fashion and written in standard English?

Reviewer #1: Yes

Reviewer #2: Yes

5. Review Comments to the Author

Reviewer #1: Thank you for inviting me to review this manuscript, examining psychological risk factors of disordered eating in athletes. The project tested a theoretical model proposed by Petrie and Greenleaf in a large, multi-sport and mixed gender sample using structural equation modelling. It also tested the utility of a newly formed model including the addition of social media as a risk factor. The newly revised model was then tested to see if it is equivalent across groups including gender, age, sport type and competition level.

The study found a poor fit for the original theoretical model, which improved slightly with the addition of social media. The newly revised model had a better fit for males than females, young athletes and those at a non-elite level.

Strengths of this manuscript include its large sample size, mixed gender sample, the clarity in writing and the importance of the research topic. Limitations include the use of research scales which have not been validated by past research and its cross-sectional design meaning that causation cannot be inferred (as noted by the authors).

My recommendation is to accept this manuscript for publication following minor amendments, as follows:

1) Can the authors provide more details on the studies that previously undertook structural testing of the model? For example, I think it would be helpful for the reader to know which pathways required adjustment in the study by Anderson et al. Further, in De Sousa Fortes and colleagues study, what sports did the male Brazilian athletes compete in? I think this information would help the reader to further understand the background to the present study, to then aid their interpretation of this study’s findings.

2) What led the authors to suspect that there were duplicates in the completion of the survey? Was this based on email addresses or IP addresses?

3) Further information on the demographic details would be useful. For the age range, 27+, what was the maximum age recruited in the study? What was the mean BMI for the study? I also think it is important to include the total N for each of the different sports that the athletes took part in so that future research could replicate this study.

4) In the limitations it should also be mentioned that due to human error one of the questions from the restrained eating measure was omitted. Which item was omitted from the scale? Please include this information in the manuscript.

5) There are a few minor typos in the article that require editing, otherwise the article is very well written. Please amend: line 57: ‘disordered eating in’; line 235 there is a % sign missing; line 226, please remove either 1,017 or N=1017 as repetition is not needed; line 357 ‘due to a lack of fit’; line 186 a full stop is missing; line 92 please spell out the acronym NCAA in full.

6) The discussion section of the article would benefit from elaboration. For instance, drawing on psychological theory why did the model fit better for certain groups and how could the findings be taken into account when considering the prevention and treatment of disordered eating in athletes?

Reviewer #2: This study entitled “Psychological Risk Indicators of Disordered Eating in Athletes” aimed at assessing the applicability of the theoretical etiological model proposed by Petrie and Greenleaf (2007; 2012) in a large sample of athletes. Overall, the study addresses a gap in the current literature and I believe that it offers an interesting contribution, but would be strengthened by addressing the following concerns.

Introduction:

General feedback: The authors need to spend more time/space explaining why is important to investigate disordered eating and eating disorders in athletes. Although I am familiar with the literature, the authors assume a lot of knowledge about this topic from the reader. I suggest explaining why this topic is important before describe the Petrie and Greenleaf (2007; 2012) model.

1) Page 3, line 57: The sentence starting with "Participation in competitive sports has the potential to increase the risk of eating disorders and disordered eating in" do not make sense. Perhaps the authors mean: in athletes?

2) Page 3, lines 63-64: Please explain briefly the dual-pathway model.

3) Page 3, line 73: Authors should clearly state that they are going to explain all the eight factors/mediators included in the model.

4) Pages 3 and 4, lines 78-83: Why the number of years of sport participation and whether an athlete is currently competing or in his/her off-season are elements that may relate to the development of disordered eating? Other variables might have been chosen. Indeed, later (lines 90-101) authors introduced the number of hours spent training per week as a relevant factor of spot pressure. Please clarify because it is misleading.

5) Page 4, lines 84-85: Please provide examples of lean sports. Again, authors assume a lot of knowledge about this topic from the reader.

6) Page 4, line 86: Please include the bracket after the number 15.

7) Page 4, line 92: Do not use the acronym (NCAA) the first time you mentioned a word.

8) Page 5, line 113: I think authors should include also the coach as a source of societal pressure (along with teammates).

9) Page 6, lines 140-142: Authors should explain deeply the debate pertaining to which type(s) of athletes experience the most body dissatisfaction and whether they experience more dissatisfaction than the general population.

10) Page 6, lines 146-148: The sentence "Research has found that negative affect in athletes, especially constructs such as fear and guilt, influenced bulimic behaviour when tested in conjunction with increased body dissatisfaction, dietary intent, and dietary intent and that certain elements of negative affect " do not make sense. Please clarify.

11) Page 8, line 186: Please include a full stop after the word analysis.

Methods & Results:

General feedback: The authors should have assessed for the presence of psychological disorders and, specifically, for EDs (and eventually removed participants with EDs) or, at least, include the lack of assessment for psychological disorder within the Limitation section.

Indeed, the range of BMI (Table 1) was comprised between 14.1 and 32.4; both a BMI of 14.1 and 32.4 may be indicative of an ED/disordered eating. Therefore, the etiological model was tested in a sample of athletes where the prevalence of EDs and disordered eating was uncertain. Authors should include this information in the limitation section.

Furthermore, authors should include internal consistency reliabilities for the current study instead of from others' investigations (see Measures section).

1) Page 9, lines 217-219: Why 10 hours of training should be an index of significant involvement in sport activity (instead of a lower/higher number of hours)?

2) Page 9, lines 227-229: It could be useful for readers the inclusion of % for each sport included in the study.

3) Page 10, lines 253-245: It is not clear to me why authors operationalized sport pressure with participation in an individual/team sport. They did not include any reference pertaining to this topic in the introduction.

4) Page 12, line 299: Include a full stop after the question mark.

Discussion:

General feedback: The discussion largely repeats the findings of the analyses, which is helpful to the reader. However, the authors do not give enough attention to theoretical explanations of the findings (i.e. page 21, lines 466-467: please provide explanations for the invariance of the model across lean and non-lean sports). In general, the authors need to spend less time re-stating their results and more time discussing them. Why are the important? What is the next step?

Furthermore, authors should discuss the clinical implications of the current study. Why your results pertaining to the Petrie and Greenleaf (2007; 2012) model are important in terms of clinical implications?

1) Page 20, line 440: Please include a full stop.

2) Page 20, line 444: Authors should rephrase the discussion in accordance with the results they found. They should refer only to binge eating and bulimia symptoms instead of talking about disordered eating in general. See also page 21, line 488.

3) Page 21: Authors should include in the future directions section the inclusion of athletes younger than 18 yeas old, given that this population was overlook by authors but represents an high-risk population for the development of EDs.

Figures:

Please include the number and the legend for each figure.

6. PLOS authors have the option to publish the peer review history of their article (what does this mean?). If published, this will include your full peer review and any attached files.

Reviewer #1: No

Reviewer #2: No

---

## [Author Response · Author response to Decision Letter 0]

22 Mar 2020

The authors thank the reviewers for their careful consideration of this manuscript. All of the suggested changes have been incorporated into the manuscripts. Responses can be found on this document as well as in the track changes and comments into the original manuscript. 

Reviewer #1: Thank you for inviting me to review this manuscript, examining psychological risk factors of disordered eating in athletes. The project tested a theoretical model proposed by Petrie and Greenleaf in a large, multi-sport and mixed gender sample using structural equation modelling. It also tested the utility of a newly formed model including the addition of social media as a risk factor. The newly revised model was then tested to see if it is equivalent across groups including gender, age, sport type and competition level.

The study found a poor fit for the original theoretical model, which improved slightly with the addition of social media. The newly revised model had a better fit for males than females, young athletes and those at a non-elite level.

Strengths of this manuscript include its large sample size, mixed gender sample, the clarity in writing and the importance of the research topic. Limitations include the use of research scales which have not been validated by past research and its cross-sectional design meaning that causation cannot be inferred (as noted by the authors).

My recommendation is to accept this manuscript for publication following minor amendments, as follows:

1) Can the authors provide more details on the studies that previously undertook structural testing of the model? For example, I think it would be helpful for the reader to know which pathways required adjustment in the study by Anderson et al. Further, in De Sousa Fortes and colleagues study, what sports did the male Brazilian athletes compete in? I think this information would help the reader to further understand the background to the present study, to then aid their interpretation of this study’s findings.

The reviewer is thanked for their insight, additional information has been added to this section of the manuscript. 

2) What led the authors to suspect that there were duplicates in the completion of the survey? Was this based on email addresses or IP addresses?

They were removed based on email addresses and exact matching answers, this information has been added to the manuscript. The authors thank the reviewer for their attention to detail. 

3) Further information on the demographic details would be useful. For the age range, 27+, what was the maximum age recruited in the study? What was the mean BMI for the study? I also think it is important to include the total N for each of the different sports that the athletes took part in so that future research could replicate this study.

The author thanks the reviewer for these suggestions, they have now been incorporated into the manuscript. 

4) In the limitations it should also be mentioned that due to human error one of the questions from the restrained eating measure was omitted. Which item was omitted from the scale? Please include this information in the manuscript.

The reviewer is thanked for this idea and the information has been added to the manuscript. 

5) There are a few minor typos in the article that require editing, otherwise the article is very well written. Please amend: line 57: ‘disordered eating in’; line 235 there is a % sign missing; line 226, please remove either 1,017 or N=1017 as repetition is not needed; line 357 ‘due to a lack of fit’; line 186 a full stop is missing; line 92 please spell out the acronym NCAA in full.

The reviewer is thanked for their spectacular attention to detail. All of these changes have been made to the manuscript. 

6) The discussion section of the article would benefit from elaboration. For instance, drawing on psychological theory why did the model fit better for certain groups and how could the findings be taken into account when considering the prevention and treatment of disordered eating in athletes?

The reviewer is thanked for this contribution. Additional information has now been added to manuscript to address this highlighted issue. The additions can be found throughout the discussion section. 

This study entitled “Psychological Risk Indicators of Disordered Eating in Athletes” aimed at assessing the applicability of the theoretical etiological model proposed by Petrie and Greenleaf (2007; 2012) in a large sample of athletes. Overall, the study addresses a gap in the current literature and I believe that it offers an interesting contribution, but would be strengthened by addressing the following concerns.

Introduction:

Reviewer #2 General feedback: The authors need to spend more time/space explaining why is important to investigate disordered eating and eating disorders in athletes. Although I am familiar with the literature, the authors assume a lot of knowledge about this topic from the reader. I suggest explaining why this topic is important before describe the Petrie and Greenleaf (2007; 2012) model.

The reviewer is thanked for this feedback, additional background information has now been included in the early stages of the manuscript. 

1) Page 3, line 57: The sentence starting with "Participation in competitive sports has the potential to increase the risk of eating disorders and disordered eating in" do not make sense. Perhaps the authors mean: in athletes?

The reviewer is thanked for their attention to detail, this mistake has been rectified. 

2) Page 3, lines 63-64: Please explain briefly the dual-pathway model.

The reviewer is thanked for this idea, more explanation has been added to the manuscript. 

3) Page 3, line 73: Authors should clearly state that they are going to explain all the eight factors/mediators included in the model.

The reviewer is thanked for the insight. This has change is now included in the manuscript. 

4) Pages 3 and 4, lines 78-83: Why the number of years of sport participation and whether an athlete is currently competing or in his/her off-season are elements that may relate to the development of disordered eating? Other variables might have been chosen. Indeed, later (lines 90-101) authors introduced the number of hours spent training per week as a relevant factor of spot pressure. Please clarify because it is misleading.

The reviewer is thanked for highlighting this issue, additional information and therefore clarification has been added throughout this section. 

5) Page 4, lines 84-85: Please provide examples of lean sports. Again, authors assume a lot of knowledge about this topic from the reader.

The reviewer is thanked for their insight, relevant examples have been added.

6) Page 4, line 86: Please include the bracket after the number 15.

Thank you for this mention, a bracket has been added. 

7) Page 4, line 92: Do not use the acronym (NCAA) the first time you mentioned a word.

This acronym has now been spelled out, many thanks for the identification of the error. 

8) Page 5, line 113: I think authors should include also the coach as a source of societal pressure (along with teammates).

The reviewer is thanked for this insight. While the research does not have the scope to touch on the role of the coach, it is an insightful idea for future work. 

9) Page 6, lines 140-142: Authors should explain deeply the debate pertaining to which type(s) of athletes experience the most body dissatisfaction and whether they experience more dissatisfaction than the general population.

The authors thank the reviewer for their insight. Additional information has been added to the manuscript to satisfy this comment. 

10) Page 6, lines 146-148: The sentence "Research has found that negative affect in athletes, especially constructs such as fear and guilt, influenced bulimic behaviour when tested in conjunction with increased body dissatisfaction, dietary intent, and dietary intent and that certain elements of negative affect " do not make sense. Please clarify.

The reviewer is thanked for this remark, the sentence has been clarified. 

11) Page 8, line 186: Please include a full stop after the word analysis.

Thank you for your attention to detail, a full stop has been added. 

Methods & Results:

General feedback: The authors should have assessed for the presence of psychological disorders and, specifically, for EDs (and eventually removed participants with EDs) or, at least, include the lack of assessment for psychological disorder within the Limitation section.

Indeed, the range of BMI (Table 1) was comprised between 14.1 and 32.4; both a BMI of 14.1 and 32.4 may be indicative of an ED/disordered eating. Therefore, the etiological model was tested in a sample of athletes where the prevalence of EDs and disordered eating was uncertain. Authors should include this information in the limitation section. Furthermore, authors should include internal consistency reliabilities for the current study instead of from others' investigations (see Measures section).

The reviewer is thanked for the detailed and helpful insight. More information on the EDE-Q global scores have been added to both the results section and limitations. Additionally, reliability calculations have been added. 

1) Page 9, lines 217-219: Why 10 hours of training should be an index of significant involvement in sport activity (instead of a lower/higher number of hours)?

The reviewer is thanked for their question, additional information has been provided in the manuscript. See page 10 for the additions. 

2) Page 9, lines 227-229: It could be useful for readers the inclusion of % for each sport included in the study.

The reviewer is thanked for this suggestion. Additional information on the percentages of each sport included in the study have been added to the manuscript. Please see page 10 for additions. 

3) Page 10, lines 253-245: It is not clear to me why authors operationalized sport pressure with participation in an individual/team sport. They did not include any reference pertaining to this topic in the introduction.

The reviewer is thanked for this remark as it is a mistake that this operationalisation was included as it was not part of the analysis. 

4) Page 12, line 299: Include a full stop after the question mark.

The reviewer is thanked for their attention to detail, this omission has been rectified. 

Discussion:

General feedback: The discussion largely repeats the findings of the analyses, which is helpful to the reader. However, the authors do not give enough attention to theoretical explanations of the findings (i.e. page 21, lines 466-467: please provide explanations for the invariance of the model across lean and non-lean sports). In general, the authors need to spend less time re-stating their results and more time discussing them. Why are the important? What is the next step?

Furthermore, authors should discuss the clinical implications of the current study. Why your results pertaining to the Petrie and Greenleaf (2007; 2012) model are important in terms of clinical implications?

The author is thanking the reviewer for this recommendation. It has been fully notes and several additional sentences have been added to the discussion of the manuscript. 

1) Page 20, line 440: Please include a full stop.

The reviewer is thanked for their attention to detail, this omission has been rectified. 

2) Page 20, line 444: Authors should rephrase the discussion in accordance with the results they found. They should refer only to binge eating and bulimia symptoms instead of talking about disordered eating in general. See also page 21, line 488.

The author thanks the reviewer for this suggestion, it has been noted and carefully implemented throughout the discussion. 

3) Page 21: Authors should include in the future directions section the inclusion of athletes younger than 18 yeas old, given that this population was overlook by authors but represents an high-risk population for the development of EDs.

The reviewer is thanked for their insight, this recommendation has been added to the manuscript. 

Figures:

Please include the number and the legend for each figure. 

The Reviewers are thanked for this comment, number and legends have been added to the figures.

---

## [Decision Letter · Decision Letter 1]

27 Apr 2020

Psychological Risk Indicators of Disordered Eating in Athletes

PONE-D-19-34507R1

Dear Mrs Stoyel

We are pleased to inform you that your manuscript has been judged scientifically suitable for publication and will be formally accepted for publication once it complies with all outstanding technical requirements.

With kind regards,

Valentina Cardi

Academic Editor

PLOS ONE

Additional Editor Comments (optional):

Both reviewers found your revisions satisfactory and have approved publication of the manuscript.

Reviewers' comments:

Reviewer's Responses to Questions

Reviewer #1: All comments have been addressed

Reviewer #2: All comments have been addressed

2. Is the manuscript technically sound, and do the data support the conclusions?

Reviewer #1: (No Response)

Reviewer #2: Yes

3. Has the statistical analysis been performed appropriately and rigorously? 

Reviewer #1: (No Response)

Reviewer #2: Yes

4. Have the authors made all data underlying the findings in their manuscript fully available?

Reviewer #1: (No Response)

Reviewer #2: No

5. Is the manuscript presented in an intelligible fashion and written in standard English?

Reviewer #1: (No Response)

Reviewer #2: Yes

6. Review Comments to the Author

Reviewer #1: (No Response)

Reviewer #2: This is a revised version of a manuscript that I previously reviewed. The authors have adequately addressed my prior concerns.

7. PLOS authors have the option to publish the peer review history of their article (what does this mean?). If published, this will include your full peer review and any attached files.

Reviewer #1: No

Reviewer #2: No

---

## [Editor Report · Acceptance letter]

1 May 2020

PONE-D-19-34507R1 

Psychological Risk Indicators of Disordered Eating in Athletes 

Dear Dr. Stoyel:

I am pleased to inform you that your manuscript has been deemed suitable for publication in PLOS ONE. Congratulations! Your manuscript is now with our production department. 

With kind regards,

on behalf of

Dr. Valentina Cardi 

Academic Editor

PLOS ONE